# Balloon-expandable versus self-expanding transcatheter aortic valve replacement for bioprosthetic dysfunction: A systematic review and meta-analysis

Hsiu-An Lee [1,2], An-Hsun Chou[3], Victor Chien-Chia Wu[4], Dong-Yi Chen[4], Hsin-Fu Lee[4], Kuang-Tso Lee[4], Pao-Hsien Chu[4], Yu-Ting Cheng[1], Shang-Hung Chang[4], Shao-Wei Chen[1,5]*

1 Division of Thoracic and Cardiovascular Surgery, Department of Surgery, Chang Gung Memorial Hospital, Linkou Medical Center, Chang Gung University, Taoyuan City, Taiwan, 2 Division of Cardiovascular Surgery, Department of Surgery, Chang Bing Show Chwan Memorial Hospital, Changhua, Taiwan, 3 Department of Anesthesiology, Chang Gung Memorial Hospital, Linkou Medical Center, Chang Gung University, Taoyuan City, Taiwan, 4 Department of Cardiology, Chang Gung Memorial Hospital, Linkou Medical Center, Chang Gung University, Taoyuan City, Taiwan, 5 Center for Big Data Analytics and Statistics, Chang Gung Memorial Hospital, Linkou Medical Center, Taoyuan City, Taiwan

* josephchen0314@gmail.com

**Data Availability Statement:** All relevant data are within the paper and its Supporting Information files.

## Abstract

### Background

Transcatheter aortic valve-in-valve (VIV) procedure is a safe alternative to conventional reoperation for bioprosthetic dysfunction. Balloon-expandable valve (BEV) and self-expanding valve (SEV) are the 2 major types of devices used. Evidence regarding the comparison of the 2 valves remains scarce.

### Methods

A systematic review and meta-analysis was conducted to compare the outcomes of BEV and SEV in transcatheter VIV for aortic bioprostheses dysfunction. A computerized search of Medline, PubMed, Embase, and Cochrane databases was performed. English-language journal articles reporting SEV or BEV outcomes of at least 10 patients were included.

### Results

In total, 27 studies were included, with 2,269 and 1,671 patients in the BEV and SEV groups, respectively. Rates of 30-day mortality and stroke did not differ significantly between the 2 groups. However, BEV was associated with significantly lower rates of postprocedural permanent pacemaker implantation (3.8% vs. 12%; $P < 0.001$). Regarding echocardiographic parameters, SEV was associated with larger postprocedural effective orifice area at 30 days (1.53 cm$^2$ vs. 1.23 cm$^2$; $P < 0.001$) and 1 year (1.55 cm$^2$ vs. 1.22 cm$^2$; $P < 0.001$).

**Funding:** This work was supported by a grant from Chang Gung Memorial Hospital, Taiwan CMRPG3H1511 (SWC), CMRPG 3J0661 (SWC), and BMRPD95 (SWC). This work was also supported by Ministry of Science and Technology grants MOST 107-2314-B-182A-152 and MOST 108-2314-B-182A-141 (SWC). The authors are thankful for the statistical assistance provided by and acknowledge the support of the Maintenance Project of the Center for Big Data Analytics and Statistics (grant CLRPG3D0045) at Chang Gung Memorial Hospital for statistical consultation and data analysis. The funders had no role in study design, data collection and analysis, decision to publish, or preparation of the manuscript.

**Competing interests:** The authors have declared that no competing interests exist.

## Conclusions

For patients who underwent transcatheter aortic VIV, SEV was associated with larger post-procedural effective orifice area but higher rates of permanent pacemaker implantation. These findings provide valuable information for optimizing device selection for transcatheter aortic VIV.

## 1. Introduction

The use of bioprosthetic valves in surgical aortic valve replacement (AVR) has increased considerably during the last few decades [1], particularly in middle-aged patients, largely driven by patients' wish of avoiding lifelong anticoagulation. However, bioprosthesis degenerates, requiring reoperation, which remains a relatively high risk. The evolution of transcatheter aortic valve replacement (TAVR) has enabled a safe and feasible alternative, the transcatheter valve-in-valve (VIV) procedure, which is less invasive than conventional redo surgery and has comparable outcomes [2–6]. Considering the possibility of future transcatheter VIV, the trend of increasing use of bioprostheses in surgical AVR is likely to persist, and the need of aortic VIV is expected to grow exponentially in the future.

Balloon-expandable valve (BEV) and self-expanding valve (SEV) are the two major types of transcatheter heart valves (THVs). These two THV types are different in valve height, implantation depth, relative position of the valve and the annulus, radial force, deployment mechanism, and valve geometry and therefore may result in different outcomes and rates of complication, such as postprocedural transvalvular pressure gradient, conduction block, or paravalvular leak (PVL). Currently, there is no randomized study comparing the two types of THVs, and only few observational studies have been published, with the observation that SEV was associated with better postprocedural hemodynamic performance but higher rates of postprocedural permanent pacemaker (PPM) implantation and aortic regurgitation [7, 8]. However, the most recent publication is a single-center study with limited number of patients and thus may not represent the whole population well [7]. Large cohort studies exist but are relatively outdated [8, 9]. Hence, a meta-analysis of the most recent studies is warranted to guide physicians in selecting the optimal device for VIV candidates.

## 2. Material and methods

We conducted this systematic review and meta-analysis in accordance with the Preferred Reporting Items for Systematic Reviews and Meta-Analyses (PRISMA) guidelines. A PRISMA checklist used for this review is provided in the **S1 Table**. The study has been registered on PROSPERO (CRD42018111178).

### 2.1. Literature search

We performed a computer search of the Medline, PubMed, Embase, and Cochrane databases using the following keywords: "transcatheter", "aortic", "valve", "failed", "failing", "degenerated", "degeneration", "degenerative", "deterioration", and "valve in valve". The detailed search strategy is provided in the **S1 Appendix**. All relevant studies published until April 2020 were identified. Review articles and meta-analyses were screened for additional studies from the cited references. The processes of searching and reviewing were independently performed by 2 evaluators (H.-A. Lee and S.-W. Chen). Discrepancies were discussed to achieve a consensus.

## 2.2. Study selection

Inclusion criteria were as follows: (1) original article with full-length content available in English, (2) at least 10 patients who underwent aortic VIV procedures for failed surgical aortic bioprosthesis using either Edwards Lifesciences or Medtronic THVs were enrolled, and (3) results of patients who underwent aortic VIV procedures with BEV or SEV were reported. Studies were excluded if they met any of the following conditions: (1) study population over-lapped with another study, including subgroup studies of a main study; (2) devices other than Medtronic valves (Medtronic, Minneapolis, MN) and Edwards Lifesciences valves (Edwards Lifesciences, Irvine, CA) were used; and (3) VIV for failed THVs. If studies were suspected of involving an overlapping cohort, only data of the most recent publication were included for analysis.

## 2.3. Data extraction

Data extracted were characteristics of the enrolled studies and characteristics of patients reported, including baseline information and outcomes. Study-level characteristics included year of publication, study period, location of the study conducted, number of hospitals, and number of patients included. Baseline patient-level information included age, Society of Thoracic Surgery (STS) score, European System for Cardiac Operative Risk Evaluation (Euro-SCORE) II, logistic EuroSCORE, comorbidities, left ventricular ejection fraction, devices used, and characteristics of previous bioprosthesis. Thirty-day and 1-year outcomes were extracted, including death of any cause, cardiovascular death, stroke, coronary artery obstruction, major vascular complications, PPM implantation, major or life-threatening bleeding, acute kidney injury, second valve required, conversion to traditional surgery, and hemodynamics of the implanted valves.

## 2.4. Quality assessment

The Newcastle–Ottawa Scale (NOS) [10] was used to assess the quality of included studies, with scores ranging from 0 (lowest quality) to 8 (highest quality). Two reviewers (H.-A. Lee and S.-W. Chen) assessed the scores of each study separately; disagreements between the 2 reviewers were discussed until a consensus was achieved.

## 2.5. Statistical analysis

The estimates of primary and secondary outcomes derived from individual studies for each arm (Medtronic or Edwards Lifesciences valves) were pooled using the random-effects model. In contrast to the fixed-effects model, a random-effects model enables the true underlying effect to vary among individual studies. $I^2$ values >25%, >50%, and >75% were considered to represent low, moderate, and high heterogeneity across the studies, respectively [11]. The pooled estimates between the BEV and SEV were compared using the mixed-effects model. In a further subgroup analysis, we compared outcomes between the Evolut R (Medtronic) and Sapien 3 (Edwards Lifesciences) valves. Statistical significance was set at $P < 0.05$ with a two-tailed test. Data were analyzed using the software Comprehensive Meta-Analysis (version 3.3; Biostat, Inc., Englewood, NJ, USA).

## 3. Results

### 3.1. Literature search

The literature screened, excluded, reviewed, and included for analysis is illustrated in **Fig 1**. Of the 398 articles yielded by computer search, 293 were excluded after titles and abstracts were

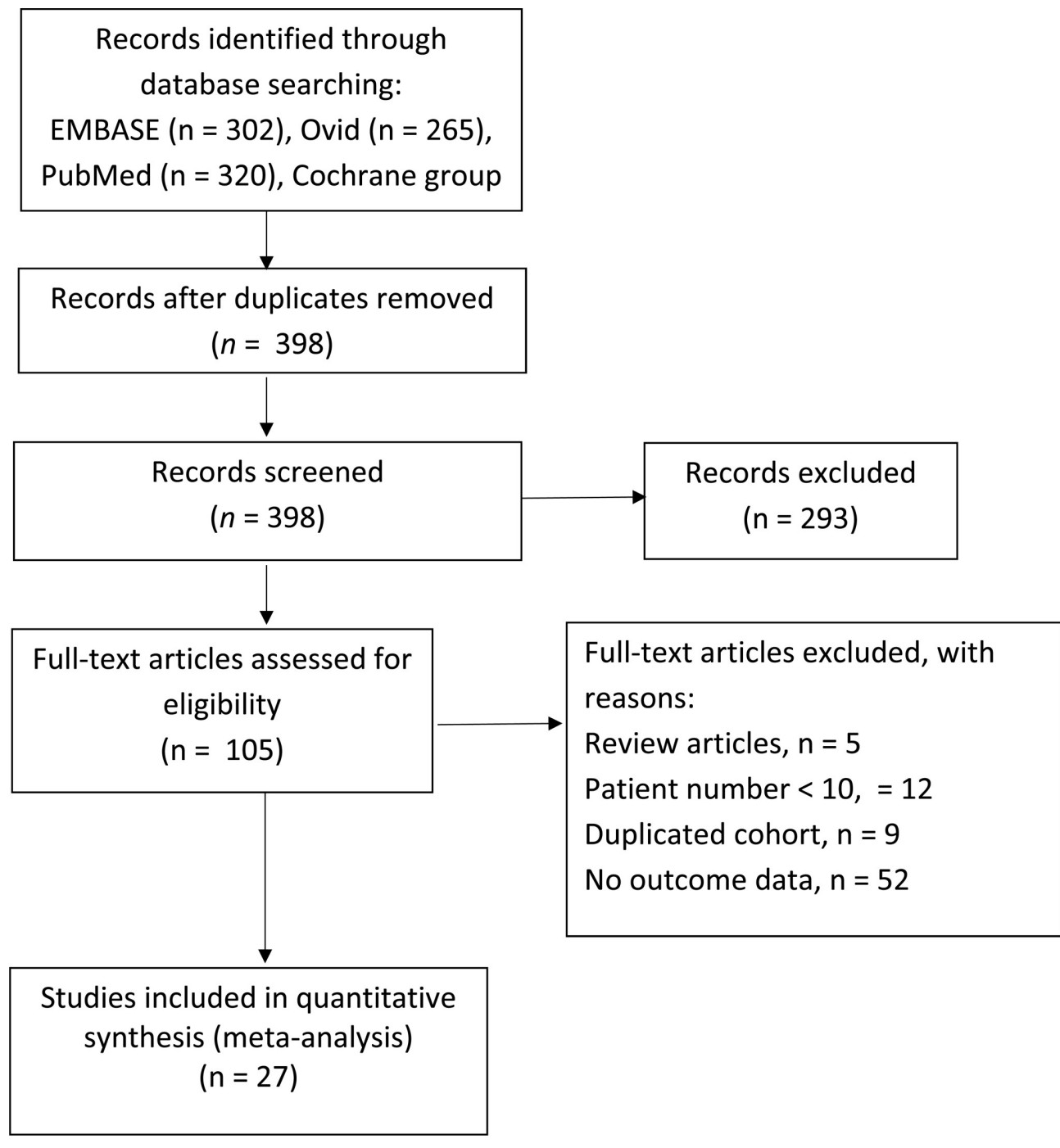

**Fig 1. Flow diagram depicting study selection process.**

screened. Full texts of 105 articles were reviewed to evaluate eligibility; of them, 5 were excluded because they were meta-analysis or review articles, 12 because their case numbers were <10, 9 because they included duplicated cohorts, and 52 because they did not report outcomes of patients who underwent VIV with BEV or SEV. Hence, 27 studies were included for the final quantitative meta-analysis [7, 8, 12–36]. All 27 studies were observational. Five of the

**Table 1. Study data.**

| First author | Year | Valve types | Study type | Locations/country | No. of centers | study period | Patient number |
|---|---|---|---|---|---|---|---|
| Woitek [34] | 2020 | BEV, SEV | Single center | Germany | 1 | 2006–2017 | 146 |
| Ribeiro [36] | 2018 | BEV, SEV | Multi-center | Global | 135 | 2007–2014 | 1324 |
| Ochiai [7] | 2018 | BEV, SEV | Single center | California, USA | 1 | 2012–2017 | 74 |
| Dvir [8] | 2014 | BEV, SEV | Multi-center | Global | 55 | 2007–2013 | 459 |
| Ihlberg [22] | 2013 | BEV, SEV | Multi-center | Nordic | 11 | 2008–2012 | 45 |
| Stankowski [29] | 2020 | SEV | Single center | Germany | 1 | 2003–2018 | 68 |
| Pascual [33] | 2019 | SEV | Single center | Spain | 1 | 2012–2017 | 45 |
| Schwerg [13] | 2018 | SEV | Single center | Germany | 1 | 2013–2017 | 26 |
| Scholtz [14] | 2018 | SEV | Single center | Germany | 1 | 2009–2016 | 37 |
| Sang [31] | 2018 | SEV | Single center | Michigan, USA | 1 | 2014–1016 | 22 |
| Deeb [17] | 2017 | SEV | Multi-center | USA | NA | 2013–2015 | 227 |
| Chhatriwalla [18] | 2017 | SEV | Single center | Michigan, USA | 9 | NA | 12 |
| Duncan [20] | 2015 | SEV | Single center | UK | 1 | 2009–2014 | 22 |
| Ong [23] | 2012 | SEV | Multi-center | Germany | 3 | NA | 18 |
| Linke [24] | 2012 | SEV | Single center | Germany | 1 | NA | 27 |
| Bedogni [27] | 2011 | SEV | Multi-center | Italy | 8 | NA | 25 |
| Murdoch [30] | 2020 | BEV | Multi-center | Global | 46 | 2012–2015 | 339 |
| Stankowski [32] | 2019 | BEV | Single center | Germany | 1 | 2010–2018 | 27 |
| Seiffert [12] | 2018 | BEV | Multi-center | Global | NA | NA | 514 |
| Webb [15] | 2017 | BEV | Multi-center | Worldwide | 34 | 2012–2014 | 365 |
| Nielsen-Kudsk [16] | 2017 | BEV | Single center | Denmark | 1 | 2015–2017 | 10 |
| Ye [19] | 2015 | BEV | Single center | Canada | 1 | 2007–2013 | 42 |
| Bapat [21] | 2014 | BEV | Single center | UK | 1 | 2010–2014 | 10 |
| Seiffert [35] | 2012 | BEV | Single center | Germany | 1 | 2008–2011 | 11 |
| Bapat [25] | 2012 | BEV | Single center | UK | 1 | 2009–2011 | 23 |
| Pasic [26] | 2011 | BEV | Single center | Germany | 1 | NA | 14 |
| Kempfert [28] | 2010 | BEV | Single center | Germany | 1 | 2007–2009 | 11 |

Basic information of studies included in the meta-analysis.

BEV, balloon-expandable valve; SEV, self-expanding valve

studies reported outcomes of both BEV and SEV, while the other 22 studies enrolled only 1 of the 2 types of THV. Basic information of the 27 studies is shown in **Table 1**. Three studies derived from Valve-In-Valve International Database were included because each of them has data that was not reported in the other articles. For items that were reported by more than 1 of the studies, only those reported by the latest publication were included in our analysis. Quality assessment was performed using the NOS, with scores of the 27 studies ranging 5–9 points (**S2 Table**).

### 3.2. Baseline and procedural characteristics

**Table 2** shows the pooled baseline and procedural characteristics of all patients in the 27 included studies. A total of 2,269 and 1,671 patients in the BEV and SEV groups were included. Mean age (78.0 ± 1.6 years in BEV vs. 75.6 ± 10.0 years in SEV), STS score (9.0 ± 2.5 in BEV vs. 9.2 ± 2.2 in SEV), left ventricular ejection fraction (50.0 ± 2.7% in BEV vs. 51.1 ± 3.0% in SEV), and other baseline echocardiographic parameters appeared to be similar between the 2 groups. The proportion of small degenerated surgical bioprostheses ($\leq$21mm) appeared slightly lower in BEV (25.6%) than in SEV (30.7%) groups; however, the proportion of small

**Table 2. Baseline and procedural characteristics of patients (number of included studies = 27).**

| Variable | BEV (Edwards) | | SEV (Medtronic) | |
|---|---|---|---|---|
| | Available data, n | Weighted % or mean ± SD | Available data, n | Weighted % or mean ± SD |
| Age (year) | 1087 | 78.0 ± 1.6 | 788 | 75.6 ± 10.0 |
| Male (%) | 1097 | 63.3% | 751 | 55.3% |
| Log EuroSCORE | 730 | 21.7 ± 9.9 | 586 | 26.2 ± 3.4 |
| EuroSCORE II | 53 | 19.6 ± 5.7 | 188 | 11.3 ± 2.9 |
| STS score | 1087 | 9.0 ± 2.5 | 678 | 9.2 ± 2.2 |
| CAD (%) | 491 | 65.2% | 431 | 61.7% |
| Prior stroke (%) | 1065 | 13.7% | 666 | 11.7% |
| Prior Afib (%) | 392 | 46.2% | 467 | 42.0% |
| Prior PPM (%) | 943 | 17.2% | 377 | 22.5% |
| PAD (%) | 1050 | 23.7% | 674 | 22.7% |
| CKD (%) | 1009 | 37.4% | 674 | 34.3% |
| AR ≥moderate (%) | 648 | 43.4% | 509 | 54.0% |
| Bioprosthesis age (year) | 732 | 10.5 ± 1.6 | 657 | 9.9 ± 1.3 |
| Stented valve (%) | 1078 | 84.2% | 867 | 75.1% |
| Stentless valve (%) | 809 | 13.6% | 867 | 19.6% |
| Bioprosthesis size (%) | 1097 | | 746 | |
| ≤21 mm | | 25.6% | | 30.7% |
| 21–24.9 mm | | 40.4% | | 37.9% |
| ≥25 mm | | 31.4% | | 28.9% |
| Unknown | | 2.3% | | 3.2% |
| Mode of failure (%) | | | | |
| AS | 1126 | 45.8% | 632 | 53.2% |
| AR | 1126 | 28.0% | 620 | 27.3% |
| Mix | 1116 | 26.6% | 583 | 21.8% |
| LVEF (%) | 829 | 50.0 ± 2.7 | 527 | 51.1 ± 3.0 |
| AV area (cm$^2$) | 699 | 0.90 ± 0.08 | 671 | 0.95 ± 0.09 |
| AVA index (cm$^2$/m$^2$) | 900 | 0.53 ± 0.06 | 213 | 0.55 |
| Max PG (mmHg) | 341 | 62.6 ± 8.5 | 423 | 61.0 ± 9.2 |
| Mean PG (mmHg) | 1022 | 34.6 ± 3.8 | 754 | 36.0 ± 4.4 |
| Fluoroscopic time (min) | 436 | 18.3 ± 3.4 | 92 | 19.6 ± 9.8 |
| THV size ≤23 mm (%) | 1299 | 67.5% | 490 | 26.9% |
| TF access (%) | 1076 | 68.5% | 561 | 95.0% |

Abbreviations: Afib, atrial fibrillation; AR, aortic regurgitation; AS, aortic stenosis; AV, aortic valve; AVA, aortic valve area; CAD, coronary artery disease; CKD, chronic kidney disease; LVEF, left ventricular ejection fraction; PAD, peripheral artery disease; PG, pressure gradient; PPM, permanent pacemaker; PVL, paravalvular leak; STS, Society of Thoracic Surgery; TF, transfemoral; THV, transcatheter heart valve.

THVs (≤23 mm) used was much higher in the BEV group (67.5%) than in the SEV group (26.9%). Transfemoral access was more frequently used in the SEV group (95%) than in the BEV group (61.3%).

### 3.3. Clinical and echocardiographic outcomes

The event rates of all-cause mortality, cardiovascular death, and stroke at 30 days did not differ significantly between the BEV and SEV groups (Fig 2A). However, BEV was associated with significantly lower rates of major vascular complications (4.7% vs. 8.7%; $P = 0.012$), PPM implantation (3.8% vs. 12%; $P < 0.001$), and second valve requirement (2.9% vs. 6.2%;

**Fig 2.** Forest plot comparing 30-day (A) and 1-year (B) clinical outcomes between BEV and SEV. BEV = balloon-expandable valve; SEV = self-expanding valve.

| A. 30-day clinical outcomes | No. of study | | Event/Total | Percentage (95% CI) | $I^2$(%) | P |
|---|---|---|---|---|---|---|
| Any-cause death | | | | | | 0.117 |
| BEV | 12 | | 19/727 | 3.6 (2.4, 5.4) | 0.0 | |
| SEV | 14 | | 33/788 | 5.5 (4.0, 7.5) | 0.0 | |
| Cardiovascular death | | | | | | 0.789 |
| BEV | 7 | | 28/683 | 4.5 (2.6, 7.9) | 29.3 | |
| SEV | 6 | | 19/551 | 4.1 (2.7, 6.3) | 0.0 | |
| Stroke | | | | | | 0.907 |
| BEV | 8 | | 14/497 | 3.3 (2.0, 5.4) | 0.0 | |
| SEV | 9 | | 11/492 | 3.5 (1.9, 6.2) | 9.4 | |
| Coronary artery obstruction | | | | | | 0.526 |
| BEV | 7 | | 18/1048 | 4.6 (3.2, 6.6) | 0.0 | |
| SEV | 9 | | 22/986 | 8.2 (6.3, 10.7) | 0.0 | |
| Major vascular complication | | | | | | 0.012 |
| BEV | 6 | | 27/662 | 4.7 (3.2, 6.7) | 0.0 | |
| SEV | 8 | | 48/639 | 8.7 (6.6, 11.3) | 0.0 | |
| Permanent pacemaker implantation | | | | | | <0.001 |
| BEV | 11 | | 25/818 | 3.8 (2.6, 5.5) | 0.0 | |
| SEV | 14 | | 86/788 | 12.0 (8.4, 16.9) | 55.7 | |
| Major or life-threatening bleeding | | | | | | 0.878 |
| BEV | 5 | | 84/615 | 5.9 (1.7, 18.4) | 85.9 | |
| SEV | 6 | | 58/539 | 6.6 (2.4, 16.7) | 81.5 | |
| Acute kidney injury | | | | | | 0.072 |
| BEV | 2 | | 29/402 | 7.2 (5.1, 10.2) | 0.0 | |
| SEV | 2 | | 10/264 | 3.8 (2.1, 6.9) | 0.0 | |
| Acute kidney injury stage 2 or 3 | | | | | | 0.329 |
| BEV | 3 | | 32/536 | 4.1 (0.6, 21.9) | 5.6 | |
| SEV | 1 | | 9/213 | 7.4 (1.9, 25.2) | 0.0 | |
| Device unsuccess | | | | | | 0.346 |
| BEV | 2 | | 2/60 | 3.3 (12.4, 0.8) | 0.0 | |
| SEV | 6 | | 19/355 | 6.6 (10.0, 4.3) | 0.0 | |
| Second valve required | | | | | | 0.004 |
| BEV | 8 | | 21/885 | 2.9 (1.9, 4.4) | 0.0 | |
| SEV | 9 | | 34/598 | 6.2 (4.5, 8.6) | 0.0 | |
| Conversion to traditional surgery | | | | | | 0.347 |
| BEV | 4 | | 5/467 | 1.6 (0.6, 4.1) | 16.7 | |
| SEV | 5 | | 1/103 | 3.2 (1.0, 9.6) | 0.0 | |

Percentage (95% CI) — axis: 0 10 20 30

| B. 1-year clinical outcomes | No. of study | | Event/Total | Percentage (95% CI) | $I^2$(%) | P |
|---|---|---|---|---|---|---|
| Any-cause death | | | | | | 0.505 |
| BEV | 4 | | 90/649 | 15.6 (11.1, 21.5) | 52.0 | |
| SEV | 5 | | 60/502 | 13.3 (9.3, 18.6) | 35.6 | |
| Stroke | | | | | | 0.937 |
| BEV | 2 | | 16/392 | 4.3 (2.6, 6.8) | 0.0 | |
| SEV | 2 | | 7/245 | 4.6 (0.9, 20.2) | 73.9 | |

Percentage (95% CI) — axis: 0 10 20 30

$P = 0.004$). One-year all-cause mortality and stroke rates were similar between the 2 groups (**Fig 2B**).

Regarding echocardiographic outcomes, SEV was associated with better hemodynamic performance than BEV, with significantly larger postoperative effective orifice area (EOA) at 30 days (1.53 cm² vs. 1.23 cm²; $P < 0.001$) and 1 year (1.55 cm² vs. 1.22 cm²; $P < 0.001$; **Fig 3A and 3C**) and lower maximal and mean pressure gradients at 1 year (respectively, 23.0 mm Hg vs. 33.3 mm Hg, $P = 0.001$; and 13 mm Hg vs. 18.4 mm Hg, $P = 0.002$; **Fig 3C**).

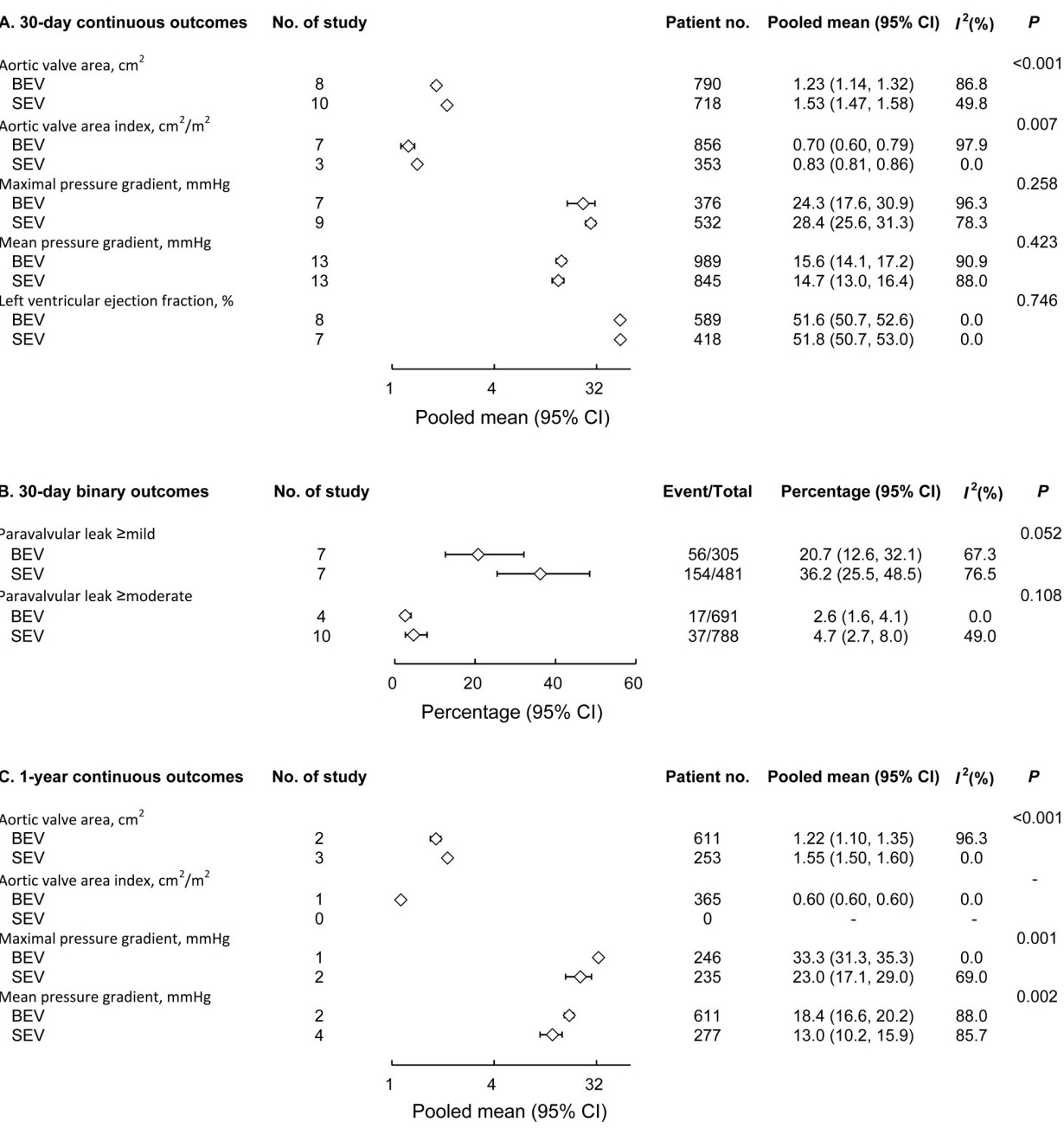

**Fig 3. Forest plot comparing echocardiographic outcomes between BEV and SEV.** Thirty-day continuous outcomes (A), 30-day binary outcomes (B), and 1-year continuous outcomes (C) of BEV and SEV were compared.

## 3.4. Subgroup analysis for newer devices

We also compared the outcomes with the Sapien 3 (Edwards Lifesciences) and Evolut R (Medtronic) valves. These are the newest generation of the 2 types of THVs with published data available for analyses. Although no statistical significance was found, Evolut R seemed to be associated with a lower mean pressure gradient than Sapien 3 (**Fig 4**).

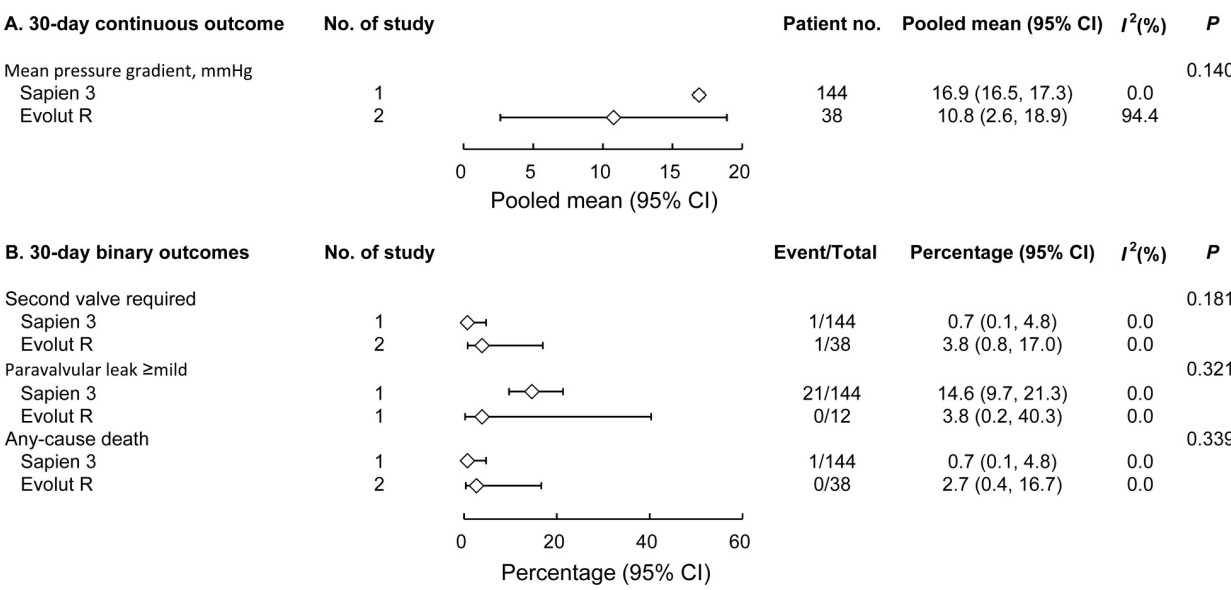

**Fig 4.** Subgroup analysis comparing 30-day outcomes of Sapien 3 and Evolut R valves for 30-day continuous outcomes (A) and 30-day binary outcomes (B).

## 4. Discussion

### 4.1. Major findings

With nearly 4000 patients included, the present meta-analysis is the largest sample used for comparing BEV and SEV outcomes in patients with failed aortic valve bioprostheses thus far. Our major findings were as follows: (1) all-cause mortality and cardiovascular death did not differ significantly between the 2 groups; (2) BEV was associated with lower rates of new PPM implantation and major vascular complications; and (3) SEV was associated with larger post-procedural EOA than BEV, both at 30 days and at 1 year.

### 4.2. New PPM implantation

SEV use is an independent risk factor for PPM implantation in the overall TAVR population [37–39]. However, previous aortic valve procedures, including surgical AVR, seemed to be protective against post-TAVR PPM implantation [37, 39], possibly because the previously implanted bioprosthesis restricted the expansion of the THV. In the present meta-analysis, the pooled PPM implantation rate after SEV implantation was 10.7%, which is nearly 3-fold that in the BEV group (3.6%; $P < 0.001$). In other words, even under the potential protection of the old bioprosthesis, SEV is still associated with significantly higher risk of postprocedural PPM implantation. This finding is consistent with previous studies focusing on aortic VIV procedure [8, 9].

### 4.3. EOA

A major concern of the aortic VIV procedure is the relatively small postprocedural aortic valve area and high transvalvular pressure gradient, mainly resulting from restricted expansion of the THVs by the old valves. Several publications, including studies using an *in vitro* model [40, 41], large cohort studies [8, 9], and propensity-matched analysis [7], reported that SEV was associated with larger postprocedural aortic valve area and lower transvalvular gradient than

BEV after aortic VIV procedures. The current meta-analysis further supported these findings in the largest sample size to date.

In the SEV we analyzed, the functioning part is positioned above the aortic annulus (i.e., the "supra-annular design," which is thought to lessen the detrimental impact on postprocedural EOA by the old valve). The theory was supported by a study using *in vitro* model in which researchers found that when the CoreValve was positioned deeper than normal, the leaflets were more constrained, and EOA decreased; and when the SAPIEN was placed more supra-annularly, the leaflets expanded more completely, and postprocedural EOA became larger [41].

One may argue that the higher percentage of small THVs (≤23mm) used in the BEV group alone can explain the smaller postprocedural EOA in BEV. However, the proportions of small degenerated surgical bioprostheses (≤21mm) were similar between the 2 groups (**Table 2**), so why were small THVs more often used in the BEV group? We believe that the supra-annular design of Medtronic SEV allows a relatively larger size, while the intra-annular design of Edward BEV results in marked leaflet distortion if the size is too large [41]. According to the ViV Aortic app, for 19 or 21mm degenerated bioprostheses, a 23mm Medtronic THV, or a 20mm Edward THV is suggested. A study using the Valve-in-Valve International Data Registry also found that elevated postprocedural pressure gradient were more common after BEV-VIV implantation than after SEV-VIV; for small surgical valves (internal diameter < 20mm) and intermediate-sized valves (internal diameter ≥20mm and <23mm) [8].

Insufficient EOA and elevated transvalvular pressure gradient not only diminish patients' physical activity and quality of life but also predict early structural valve degeneration in bioprosthetic heart valves [42]. In addition, incomplete THV expansion itself leads to localized high stress within the leaflets, which may accelerate valve degeneration [43].

## 4.4. SEV versus BEV

According to the present meta-analysis, SEV was associated with significantly better postprocedural EOA, which can reduce the risk of patient–prosthesis mismatch and improve quality of life, particularly in patients with larger body size or whose old bioprosthesis is small. Lower transvalvular gradient and better THV expansion may also lead to superior durability of the THV, which is important in patients with life expectancy of 20 years or longer. Nevertheless, higher EOA and lower gradient of SEV did not translate in to lower mortality. Moreover, SEV was associated with higher rates of postprocedural PPM implantation, which is detrimental to late outcome [37, 44].

Therefore, CoreValve may be beneficial in patients whose previous surgical valve is small and those at high risk of patient–prosthesis mismatch. However, Edwards valves may be preferred to Medtronic valves for patients with adequate surgical valve size, particularly those who are prone to encounter postprocedural PPM implantation or PVL, including patients who are older [37] and those who have prior conduction disturbances [38] or a prolonged PR interval [45]. For every transcatheter aortic VIV candidate, particularly younger patients, the valve selection decision should be made carefully after thorough consideration of device characteristics and patient condition and preference, as well as detailed explanation and discussion.

## 4.5. Study limitations

The study has several limitations. First, this meta-analysis was based on published articles; therefore, data quality and availability are limited. Second, owing to a lack of randomized controlled trials in this area, all studies included were observational, so our results can only be

interpreted as "associations," rather than as "causations." However, the absence of randomized studies warrants the present meta-analysis to help in optimizing device selection. Third, THV devices continue advancing rapidly, so the outcomes of the present study may differ from those of the newest device.

## 5. Conclusion

The present systematic review and meta-analysis found that for patients who underwent transcatheter aortic VIV, SEV was associated with significantly larger postprocedural EOA but higher rates of PPM implantation and PVL of moderate or higher degree. These findings provide valuable information in guiding proper management for patients with degenerated aortic bioprostheses.

## Supporting information

**S1 Appendix. Detailed search strategy.**
(DOCX)

**S1 Table. Prisma 2009 checklist.**
(DOC)

**S2 Table. Newcastle-Ottawa Scale quality assessment of included studies.**
(DOCX)

## Acknowledgments

The authors thank Alfred Hsing-Fen Lin and Zoe Ya-Jhu Syu for their assistance with the statistical analysis.

## Author Contributions

**Conceptualization:** Kuang-Tso Lee, Pao-Hsien Chu, Shao-Wei Chen.

**Data curation:** Hsiu-An Lee, Dong-Yi Chen, Hsin-Fu Lee, Shao-Wei Chen.

**Formal analysis:** Hsiu-An Lee, Hsin-Fu Lee, Yu-Ting Cheng.

**Funding acquisition:** Shao-Wei Chen.

**Investigation:** Hsiu-An Lee, Kuang-Tso Lee.

**Methodology:** An-Hsun Chou, Dong-Yi Chen, Kuang-Tso Lee, Shang-Hung Chang, Shao-Wei Chen.

**Project administration:** Hsiu-An Lee.

**Resources:** Shang-Hung Chang, Shao-Wei Chen.

**Software:** Dong-Yi Chen, Hsin-Fu Lee, Yu-Ting Cheng.

**Supervision:** An-Hsun Chou, Pao-Hsien Chu.

**Validation:** An-Hsun Chou, Pao-Hsien Chu, Shang-Hung Chang.

**Writing – original draft:** Hsiu-An Lee.

**Writing – review & editing:** Hsiu-An Lee, Victor Chien-Chia Wu, Pao-Hsien Chu.

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
