## [Decision Letter · Decision Letter 0]

31 Mar 2020

PONE-D-20-05542

Balloon-expandable versus self-expanding transcatheter aortic valve replacement for bioprosthetic dysfunction: a systematic review and meta-Analysis

PLOS ONE

Dear Dr Chen,

Thank you for submitting your manuscript to PLOS ONE. After careful consideration, we feel that it has merit but does not fully meet PLOS ONE’s publication criteria as it currently stands. Therefore, we invite you to submit a revised version of the manuscript that addresses the points raised during the review process.

We would appreciate receiving your revised manuscript by May 15 2020 11:59PM. To enhance the reproducibility of your results, we recommend that if applicable you deposit your laboratory protocols in protocols.io, where a protocol can be assigned its own identifier (DOI) such that it can be cited independently in the future. For instructions see: http://journals.plos.org/plosone/s/submission-guidelines#loc-laboratory-protocols

We look forward to receiving your revised manuscript.

Kind regards,

Corstiaan den Uil

Academic Editor

PLOS ONE

Journal Requirements:

2. We noticed that the search of your systematic review was last performed in October 2018. Please ensure that the search is up to date and that the systematic review/meta-analysis includes any new studies published since then.

Reviewers' comments:

Reviewer's Responses to Questions

**Comments to the Author**

1. Is the manuscript technically sound, and do the data support the conclusions?

Reviewer #1: Yes

Reviewer #2: Yes

2. Has the statistical analysis been performed appropriately and rigorously? 

Reviewer #1: N/A

Reviewer #2: Yes

3. Have the authors made all data underlying the findings in their manuscript fully available?

Reviewer #1: Yes

Reviewer #2: Yes

4. Is the manuscript presented in an intelligible fashion and written in standard English?

Reviewer #1: Yes

Reviewer #2: Yes

5. Review Comments to the Author

Reviewer #1: This is a very interesting review and meta-analysis of VIV-treatment. However, there are some points to discuss:

1. Comparison of EOA between SEV and BEV:

It should be better explained why the EOA of SEV is bigger than BEV. The used BEV-Size was smaller than the SEV-Sizes (THV size ≤ 23mm 65,7% vs 31,7%), this alone can explain the difference. How was the gradient-/EOA- difference in treated small surgical bioprosthesis (<21mm), are there data available?

2. There was a recent publication about measuring error of measured echo gradients in intra annular and supra annular valves (Abbas AE and Pibarot P, CCI 2019). Are there data about invasive measured gradients after VIV available.

3. For a recommendation which valve should be used in which situation, the coronary access possibility after VIV should be discussed, as there are differences between BEV and SEV.

Reviewer #2: it is a very important subject that the authors have performed a metanalysis on given the sparse RCT data comparing these 2 valves. Some questions that need to be answered:

1. Why is there such a large difference between the % of baseline >=moderate MR between the 2 groups (25.5% vs 68.3%)

2. Could the higher number of stented valves in the BEV group explain the higher PVL after the procedures?

3. Given that transfemoral access is the most common modality why were there only 67.8% in the BEV group? Especially given that alternate access is associated with higher periprocedural complication rates.

4. Higher EOA and Lower gradients have been shown in prior studies with SEV, although they did not translate into lower mortality with SEV compared to BEV. Can the authors elaborate on that in their discussion?

6. PLOS authors have the option to publish the peer review history of their article (what does this mean?). If published, this will include your full peer review and any attached files.

Reviewer #1: No

Reviewer #2: No

---

## [Author Response · Author response to Decision Letter 0]

14 May 2020

The following is a point-by-point response to reviewers comments.

2. We noticed that the search of your systematic review was last performed in October 2018. Please ensure that the search is up to date and that the systematic review/meta-analysis includes any new studies published since then

Response: We have updated the search to April 2020.

Reviewer #1: This is a very interesting review and meta-analysis of VIV-treatment. However, there are some points to discuss:

1. Comparison of EOA between SEV and BEV:

It should be better explained why the EOA of SEV is bigger than BEV. The used BEV-Size was smaller than the SEV-Sizes (THV size ≤ 23mm 65,7% vs 31,7%), this alone can explain the difference. How was the gradient-/EOA- difference in treated small surgical bioprosthesis (<21mm), are there data available?

Response: Thanks for the great question. Regarding gradient-/EOA- difference in treated small surgical bioprostheses, a study using the Valve-in-Valve International Data Registry found that elevated postprocedural pressure gradient (mean ≥ 20mmHg) were more common with BEVs in comparison with SEVs; for small surgical valves (internal diameter [ID] < 20mm), 41.2% vs 23.4% (P = .04) and for intermediate-sized valves (≥20 and < 23), 35.8% vs 19.4% (P = .01), respectively [1](eFigure 2C). Moreover, 11.8% of Edwards SAPIEN VIV procedures performed inside small bioprosthesis had very high postprocedural gradients (mean ≥ 40mmHg), while no cases of CoreValve VIV procedures resulted in very high gradients (P = 0.005, eFigure 2D).

We believe that the supra-annular design of Medtronic SEV allows a relatively larger size, while an intra-annular design of Edwards BEV results in marked leaflet distortion if the size is too large. This can explain why the ViV Aortic app suggests 20mm Edwards THVs inside19 or 21mm degenerated bioprosthesis, but 23mm Medtronic THVs inside same-sized bioprostheses.

In conclusion, the supra-annular design of Medtronic CoreValve allows better expansion of the leaflets, and allows a relatively larger sized THV to be inserted inside a small bioprosthesis, hence results in better EOA/gradient.

We have added the above explanation and citation in the revised manuscript.

2. There was a recent publication about measuring error of measured echo gradients in intra annular and supra annular valves (Abbas AE and Pibarot P, CCI 2019). Are there data about invasive measured gradients after VIV available.

Response: Abbas et. al. demonstrated the catheterization/echocardiography discordance after native TAVR and after Valve-in-Valve TAVR[2]. The echocardiography mean gradient is significantly higher than catheterization gradient after both procedures.

In all the studies we reviewed, only 1 reported postprocedural catheterization gradients[3]. The postprocedural mean echocardiographic gradient and mean invasive gradient were 22.8 and 13.6 mmHg, respectively (P < 0.001).

3. For a recommendation which valve should be used in which situation, the coronary access possibility after VIV should be discussed, as there are differences between BEV and SEV.

Response: Yes, the coronary access possibility after VIV is a very important concern. Allali et. al. reported their experience of coronary intervention after TAVR[4]. However, the aim of the current meta-analysis is to collect evidences and analyze. Since we did not find sufficient data regarding coronary-related outcomes after ViV to perform meta-analysis, we did not include this point in our manuscript. Nevertheless, if you consider the point suitable for our article, we would like to add it in our manuscript according to your suggestion. 

Reviewer #2: it is a very important subject that the authors have performed a metanalysis on given the sparse RCT data comparing these 2 valves. Some questions that need to be answered:

1. Why is there such a large difference between the % of baseline >=moderate MR between the 2 groups (25.5% vs 68.3%)

Response: In the original analysis, very few study reported the percentage of baseline ≥moderate AR. A single study of high or low % made a large difference in the results. After collecting more data, the % of baseline ≥moderate AR was 43.4% and 54.0 in the BEV and SEV groups, respectively. However, it should be noted that only a small portion of studies reported baseline ≥moderate AR, so the result does not well represent the whole population.

2. Could the higher number of stented valves in the BEV group explain the higher PVL after the procedures?

Response: In studies comparing outcomes following VIV inside stentless versus stented bioprostheses, stentless bioprostheses appeared to be related to higher PVL than stented bioprostheses[5, 6]. Hence, the higher percentage of stentless valves in the SEV group in our study could contribute to the higher rates of postprocedural PVL. 

However, in the revised manuscript, after updating research with more recent data included in the meta-analysis, the difference of postprocedural PVL between BEV and SEV groups was no longer significant.

3. Given that transfemoral access is the most common modality why were there only 67.8% in the BEV group? Especially given that alternate access is associated with higher periprocedural complication rates.

Response: The proportion of transfemoral (TF) access reported by the two largest registry regarding Edwards VIV, VIVID Registry and PARTNER 2 VIV Registry, were 66.7% and 75.4%, respectively[7, 8]. In VIVID Registry, the TF ratio of Sapien XT was only 58.5%. Although nowadays approximately 5% of TAVR candidates require a non-femoral access, in the earlier era, 10~20% of patients require non-femoral access because previous-generation devices had larger profile. Transapical access, which could only be performed using BEV, was the first-developed non-femoral access, and had been the most commonly used non-femoral access for quite a few years. This may explain the lower percentage of TF access in BEV group, especially in earlier era. After updating the search to April, 2020, the TF ratio of BEV group slightly increased to 68.5%

4. Higher EOA and Lower gradients have been shown in prior studies with SEV, although they did not translate into lower mortality with SEV compared to BEV. Can the authors elaborate on that in their discussion?

Response: Thanks for the suggestion. We have discussed EOA and gradients in sections 4.3 and 4.4 of discussion in the original manuscript. We also added a sentence in the first paragraph of section 4.4 to emphasize that higher EOA and lower gradient of SEV did not translate into lower mortality. If there is any further suggestion from the reviewer, we will be glad to revise our manuscript accordingly. 

References

1. Dvir D, Webb JG, Bleiziffer S, Pasic M, Waksman R, Kodali S, et al. Transcatheter aortic valve implantation in failed bioprosthetic surgical valves. JAMA 2014;312(2):162-70. doi: 10.1001/jama.2014.7246.

2. Abbas AE, Mando R, Hanzel G, Gallagher M, Safian R, Hanson I, et al. Invasive Versus Echocardiographic Evaluation of Transvalvular Gradients Immediately Post-Transcatheter Aortic Valve Replacement. Circ Cardiovasc Interv. 2019;12(7):e007973. Epub 2019/07/06. doi: 10.1161/CIRCINTERVENTIONS.119.007973. PubMed PMID: 31272227.

3. Scholtz S, Piper C, Horstkotte D, Gummert J, Ensminger SM, Borgermann J, et al. Valve-in-valve transcatheter aortic valve implantation with CoreValve/Evolut R((c)) for degenerated small versus bigger bioprostheses. J Interv Cardiol. 2018;31(3):384-90. Epub 2018/03/01. doi: 10.1111/joic.12498. PubMed PMID: 29490430.

4. Allali A, El-Mawardy M, Schwarz B, Sato T, Geist V, Toelg R, et al. Incidence, feasibility and outcome of percutaneous coronary intervention after transcatheter aortic valve implantation with a self-expanding prosthesis. Results from a single center experience. Cardiovasc Revasc Med. 2016;17(6):391-8. Epub 2016/07/12. doi: 10.1016/j.carrev.2016.05.010. PubMed PMID: 27396607.

5. Duncan A, Moat N, Simonato M, de Weger A, Kempfert J, Eggebrecht H, et al. Outcomes Following Transcatheter Aortic Valve Replacement for Degenerative Stentless Versus Stented Bioprostheses. JACC Cardiovasc Interv. 2019;12(13):1256-63. Epub 2019/06/17. doi: 10.1016/j.jcin.2019.02.036. PubMed PMID: 31202944.

6. Choi CH, Cheng V, Malaver D, Kon N, Kincaid EH, Gandhi SK, et al. A comparison of valve-in-valve transcatheter aortic valve replacement in failed stentless versus stented surgical bioprosthetic aortic valves. Catheter Cardiovasc Interv. 2019;93(6):1106-15. Epub 2018/12/28. doi: 10.1002/ccd.28039. PubMed PMID: 30588736; PubMed Central PMCID: PMCPMC6590419.

7. Seiffert M, Treede H, Schofer J, Linke A, Wöhrle J, Baumbach H, et al. Matched comparison of next- and early-generation balloonexpandable transcatheter heart valve implantations in failed surgical aortic bioprostheses. EuroIntervention. 2018;14(4):e397-e404. doi: 10.4244/EIJ-D-17-00546.

8. Webb JG, Mack MJ, White JM, Dvir D, Blanke P, Herrmann HC, et al. Transcatheter Aortic Valve Implantation Within Degenerated Aortic Surgical Bioprostheses: PARTNER 2 Valve-in-Valve Registry. J Am Coll Cardiol. 2017;69(18):2253-62. Epub 2017/05/06. doi: 10.1016/j.jacc.2017.02.057. PubMed PMID: 28473128.

---

## [Editor Report · Decision Letter 1]

15 May 2020

Balloon-expandable versus self-expanding transcatheter aortic valve replacement for bioprosthetic dysfunction: a systematic review and meta-analysis

PONE-D-20-05542R1

Dear Dr. Chen,

We are pleased to inform you that your manuscript has been judged scientifically suitable for publication and will be formally accepted for publication once it complies with all outstanding technical requirements.

With kind regards,

Corstiaan den Uil

Academic Editor

PLOS ONE
---

## [Editor Report · Acceptance letter]

22 May 2020

PONE-D-20-05542R1 

Balloon-expandable versus self-expanding transcatheter aortic valve replacement for bioprosthetic dysfunction: a systematic review and meta-analysis 

Dear Dr. Chen:

I am pleased to inform you that your manuscript has been deemed suitable for publication in PLOS ONE. Congratulations! Your manuscript is now with our production department. 

With kind regards,

on behalf of

Dr. Corstiaan den Uil 

Academic Editor

PLOS ONE